# Cleft Lip and/or Cleft Palate: Prenatal Accuracy, Postnatal Course, and Long-Term Outcomes

**DOI:** 10.3390/children9121880

**Published:** 2022-11-30

**Authors:** Sivan Farladansky-Gershnabel, Hadar Gluska, Nufar Halevi, Neta Kotser, Maya Sharon-Weiner, Hanoch Schreiber, Rivka Sukenik-Halevi, Yaniv Ebner, Shmuel Arnon, Ofer Markovitch

**Affiliations:** 1Department of Obstetrics and Gynecology, Meir Medical Center, Kfar Saba 4428163, Israel; 2Sackler School of Medicine, Tel Aviv University, Tel Aviv 6329302, Israel; 3Genetics Institute, Meir Medical Center, Kfar Saba 4428163, Israel; 4Department of Otolaryngology, Meir Medical Center, Kfar Saba 4428163, Israel; 5Department of Neonatology, Meir Medical Center, Kfar Saba 4428163, Israel

**Keywords:** cleft lip, cleft palate, prenatal diagnosis, anomaly scan, post-natal, ultrasound diagnosis

## Abstract

Orofacial clefts include cleft lip (CL) and cleft palate (CP). This retrospective study assessed the efficacy of prenatal sonographic diagnosis of isolated and non-isolated cases of CL/CP and the postnatal outcomes of these children. Data regarding patients diagnosed and treated in the tertiary orofacial clinic from 2000 to 2020 were retrieved from electronic medical records and telephone-based questionnaires. Isolated CL was found in 7 cases (7.2%), isolated CP in 51 (53%), and combined CL/CP in 38 (39.5%), and 22 cases (23%) were associated with other anomalies. Among 96 cases, 39 (40.6%) were diagnosed prenatally. Isolated CL was diagnosed in 5/7 (71.5%), combined clefts in 29/38 (76.3%), and CP in 7/51 (13.8%). Prenatal chromosomal analysis performed in 32/39 (82%) cases was normal for all. The rate of surgical intervention in the first year of life was 36/38 (94.7%) for combined clefts, 5/7 (71.4%) for CL, and 20/51 (39%) for isolated CP. Most children had speech therapy (23/38 (60.5%), 3/7 (42.8%), and 41/51 (80.3%), respectively) and psychotherapy (6/38 (15.7%), 3/7 (42.8%) and, 15/51 (29.4%), respectively). The accuracy rate of sonographic prenatal diagnosis is low. Our results emphasize the suggested work-up of fetuses with CL and/or CP and improvements to parental counseling, as well as their understanding and compliance regarding post-natal therapeutic plans.

## 1. Introduction

Cleft lip (CL) and/or cleft palate (CP) are the second most common congenital anomalies, and the most frequently detected craniofacial malformations [1]. The worldwide prevalence of orofacial clefts varies from 1 to 2.2/1000 live births [2,3]. Facial cleft is attributed to failure of the nasal and maxillofacial processes to fuse during embryogenesis at 6 to 8 weeks of embryonic life [4]. Various factors are involved in the etiology of these malformations. The most familiar triggers are genetic predispositions and teratogenic exposure during the second and third months of gestation [5,6].

CL/CP may be isolated or associated with additional congenital anomalies or other medical problems. Previous studies reported that most cases were isolated (61.6%), whereas complicated cases were less common (38.4%)^6^. Associated anomalies primarily included congenital heart defects (31.1%), hydrocephalus (11.2%), and urinary tract defects (9.7%) [7]. 

Genetic etiology is common in cases of orofacial clefts. Isolated CL/CP is mainly associated with multifactorial inheritance and increased incidence among family members [8,9,10,11,12]. Syndromic clefts are associated with variable genetic etiologies including chromosomal and monogenic syndromes. Currently, there are hundreds of genetic syndromes that have been associated with cleft lip and palate [7].

Multiple environmental factors including smoking, pregestational diabetes, alcohol abuse, certain anticonvulsants, and nutritional deficiencies have also been associated with CL/CP [8,9,10,11,12]. 

Facial clefts, among other craniofacial abnormalities, are usually identified via sonographic scan performed as part of the routine pregnancy follow-up. Previous studies reported that only 20–30% of orofacial clefts are diagnosed prenatally [13,14,15]. Nevertheless, ultrasonography is currently the routine modality for prenatal diagnosis of facial malformations [16].

Previous studies found that two-thirds of orofacial clefts involve CL and CP, and the remaining third comprises CP alone [17]. Whether an orofacial cleft is an isolated finding or is in conjunction with other anomalies is crucial for prenatal counseling, as well as for obstetrical and neonatal management. A recent study found an increased risk for still birth among fetuses with CL/P [18]. Following the diagnosis of CL and/or CP, a multidisciplinary team must provide the expectant parents with detailed information regarding the prognosis and management. Treatment programs usually involve specialists in obstetrics, genetics, neonatology, pediatrics, oral and maxillofacial surgery, plastic surgery, and otolaryngology. From birth, the cleft patient may face breathing and feeding difficulties which need which need to be addressed and dealt with (special feeding bottles, specialist consultations for right positioning, alternative feeding methods, etc). As they grow, these children are likely to require speech therapy, and sometimes psychological counseling. The prognosis differs among patients and depends on the accuracy of the prenatal diagnosis and on the presence of other anomalies. Children with secondary CP that involves the soft palate are prone to deglutition and speech disorders. Disruption of the fusion of the embryonic facial tissues that form the lip, alveolus, and primary palate may present as a cleft of the lip only, the alveolus only, or more commonly both together. The alveolar cleft often will continue either side of the primary palate to the incisive foramen. These presentations are collectively abbreviated as CL. Such presentations are typically associated with dental eruption problems necessitating orthodontic treatment [19]. Social adjustment disorders related to facial appearance are also prevalent [20].

This study assessed the accuracy of the prenatal diagnosis, as well as the work-up, postnatal outcomes, and the natural course of children diagnosed prenatally with CL and/or CP as an isolated finding or as a part of a syndrome.

## 2. Methods

This retrospective, observational study was conducted at a tertiary medical center in Israel. Data regarding all patients diagnosed and treated in the CL/CP clinic who were born from 2000 to 2020 were retrieved from the electronic medical records of children treated in a multidisciplinary orofacial anomaly clinic. A pediatric otolaryngologist (ENT) specializing in orofacial anomalies conducted the postnatal evaluations and treatment. The diagnosis was based on physical examination and imaging, and was classified using the Millard classification system into two distinct groups. The first group included class 2 and class 6 clefts, which involve only the soft tissue. These were considered incomplete clefts (abbreviated hereafter as iCL). The second group included classes 3, 4, and 7–11, which all involve the hard primary palate and/or the secondary palate (abbreviated hereafter as psCP) [3]. These groupings were made in preference to the embryonal (primary vs. secondary) classification because they separates the clinical severity to mainly aesthetic with iCL as opposed to mainly functional with psCP, and better relate to the current diagnostic limitations of ultrasonography.

Data regarding demographic characteristics, obstetrical history, sonographic findings, prenatal evaluation, pregnancy surveillance, labor and delivery outcomes, and immediate postnatal complications were retrieved from the medical records and from telephone-based questionnaires. The prenatal ultrasound exams in these cases included early anatomy scan performed at 14–16 weeks of gestation, second trimester anatomy scan performed at 21–24 weeks of gestation as part of routine pregnancy follow-up, and third trimester scan performed as needed. The exams included 2D and 3D ultrasound. Information regarding follow-up in the iCL/psCP clinic, surgical interventions, and respiratory, deglutition or speech sequelae, were also collected. Questionnaire data were verified with each patient’s medical record. All data and verbal consents were recorded and were kept confidential, to ensure the privacy of the study participants. Cases where medical data were missing or telephone questionnaire data were missing were excluded.

The parameters evaluated included clinical characteristics, prenatal ultrasound accuracy, prenatal workup, and postpartum management and outcomes. Prenatal diagnosis accuracy was further divided into complete or partial accuracy, defined as prenatal diagnosis of cleft that differed from the cleft diagnosis of the ENT specialist after birth.

Prenatal diagnosis and postnatal treatment and outcomes were compared among solely iCL, solely psCP, and combined iCL and psCP. Additional comparisons were performed between neonates with isolated iCL and/or psCP and those with iCL and/or psCP associated with additional non-orofacial abnormalities.

The outcomes of iCL and/or psCP diagnosed during pregnancy were compared with the outcomes of those diagnosed post-partum only.

### 2.1. Ethics

The study was conducted in accordance with the Declaration of Helsinki, and the protocol (Protocol number 0123-20) was approved by the Ethics Committee of Meir Medical Center. Informed consent for record review was not required as the data were obtained retrospectively. Consent for telephone interview was obtained verbally from participants who agreed to respond to the telephone questionnaire.

### 2.2. Statistical Analysis

Data were analyzed using SPSS-25 (IBM Corp., Armonk, NY, USA). Statistical significance between the two groups was calculated using the Chi-square test or Fisher’s exact test for differences in quantitative variables, and t-test or Mann–Whitney test for continuous variables, each when appropriate. Adjusted *p* < 0.05 was set as the level of statistical difference because of the performance of multiple comparisons.

## 3. Results

### 3.1. Clinical Characteristics

iCL and/or psCP were diagnosed postnatally in 96 children. Among them, iCL alone was found in 7 (7.2%), psCP alone in 51 (53%), and combined iCL and psCP in 38/96 (39.5%) cases. All cases of CL were isolated, without additional systemic anomalies. Among the 51 newborns with psCP only, 35 were isolated and 16 were not isolated, whereas in the 38 cases with combined psCP and iCL, 32 were isolated and 6 had additional abnormal findings (Figure 1).

The clinical characteristics of the entire cohort are summarized in Table 1. Medical history revealed 14/96 (14.5%) cases with a family history of iCL/psCP, 8/96 (8.3%) cases with a parental history of iCL/psCP, and 6 (6.2%) cases with an affected sibling.

Mean maternal age at birth was 29.9 years and mean gestational age at delivery was 38 weeks. Most fetuses were born vaginally (71/96, 73.9%). Most affected newborns were male (54/96, 56.2%) and 7/96 (7.2%) were one of a twin pregnancy.

For categorical variables, results are presented as *n* (%) and for continuous variables, as mean ± standard deviation (SD).

### 3.2. Accuracy of Prenatal Ultrasound Diagnosis

Among 96 newborns with iCL and/or psCP, 39 (40.6) were diagnosed prenatally. iCL alone was diagnosed in 5/7 (71.5%) cases, psCP alone in 7/51 (13.8%) cases, and combined clefts were detected in 27/38 (71%) cases. The diagnosis was verified in 3/7 (42.8%) cases of iCL, 5/51 (9.8%) cases of psCP, and 14/38 (36.8%) cases of combined cleft. Missed diagnosis occurred in 2/7 (28.6%), 2/51 (3.9%), and 13/38 (34.2%) of cases, respectively. Prenatal ultrasound failed to detect 2/7 (28.6%) cases of iCL alone, 44/51 (86.3) of psCP alone, and 11/38 of iCL and psCP combined (Table 2).

Prenatal diagnosis was made at 14–20 weeks of gestation in 19/39 (48.7%) pregnancies, at 21–24 weeks in 13/39 (33.3%), and after 24 weeks of gestation in 7/39 (18%).

Among the entire cohort, additional anomalies found during routine prenatal ultrasound included 2/96 (2%) cases of cardiac defects, 2/96 (2%) cases of CNS anomaly, 1/96 (1%) of urinary anomaly, and 6/96 (6.25%) cases of multiple anomalies.

### 3.3. Prenatal Work-Up

Fetal echocardiogram was undertaken in 34 of the 39 (87.1%) fetuses diagnosed during pregnancy. A cardiac defect was diagnosed in 4/34 (10.2%) of these cases.

Genetic counseling was performed in all cases. Amniocentesis or chorionic villus sampling was performed in 32/39 (82%) cases. Of these, 10 underwent karyotyping and 22 chromosomal microarray analysis (CMA). Genetic testing was performed in 8/22 (36.3%) of fetuses with additional anomalies. All genetic samples yielded normal results.

Prenatal fetal magnetic resonance imaging was performed in 7/39 (17.9%) cases where the diagnosis of psCP was uncertain. Prenatal ENT counseling was performed in 15/39 (38.4%) cases with suspected psCP involvement.

### 3.4. Isolated Cases vs. Associated with Additional Abnormalities

The data regarding isolated vs. non-isolated iCL/psCP are presented in Table 3. Among the 96 cases in the study group, 22 (23%) had additional anomalies. These included 4 cases with cardiac defects, 3 with CNS malformations, 3 with renal/urinary system anomalies, 5 with other facial anomalies and 7 with multiple anomalies. Among the 22 cases with additional anomalies, 14 (63.6%) were diagnosed only after birth.

Non-isolated cases had significantly higher rate of Apgar score ≤ 7 (3/22 (13.6%) vs. 0, *p* < 0.001), and immediate postnatal respiratory complications (13/22 (59.1%) vs. 8/74 (10.8%), *p* = 0.001). Non-isolated cases had significantly higher complication rates during the first year of life (Respiratory: 10/22 (45.5%) vs. 6/74 (8.1%), p < 0.001; swallowing difficulties: 15/22 (68.2%) vs. 38/74 (51.4%), p = 0.005 and speech difficulties: 11/22 (50%) vs. 14/74 (18.9%), p <0.001). They also had significantly higher rates of long-term sequelae (respiratory: 7/24 (31.8%) vs. 4/74 (5.4%), p = 0.003; swallowing difficulties: 10/22 (45.5%) vs. 10/74 (13.5%), p = 0.005 and speech difficulties: 20/22 (90.9%) vs. 43/74 (58.1%), *p* = 0.01). Non-isolated cases had a significantly higher rate of psychological therapy after the first year of life (12/22 (54.5%) vs. 19/74 (25.7%), *p* = 0.035).

There were no significant differences in demographic or obstetrical parameters between cases of isolated vs. not-isolated iCL/psCP. There was no difference regarding the laterality of the cleft between groups.

### 3.5. Postnatal Treatment

Postnatal treatment was multidisciplinary and included surgical corrective treatment, cosmetic treatment (scar revision), speech therapy, and psychotherapy.

Among the 96 cases of iCL/psCP confirmed postnatally, 61 (63.5%) infants required one or more surgical procedures during the first year of life. The rate of surgical intervention in the first year of life was 36/38 (94.7%) for those with combined clefts, 5/7 (71.4%) in cases of iCL, and 20/51 (39%) in isolated psCP. After the first year of life, surgical interventions were 23/38 (60.5%), 4/7 (57.1%), and 45/51 (88.2%), respectively. Cosmetic interventions (scar revision) were performed in 10/38 (26.3%), 3/7 (42.8%), and 2/51 (3.9%) cases, respectively. Most children had speech therapy (23/38 (60.5%), 3/7 (42.8%), and 41/51 (80.3%), respectively), as well as psychotherapy (6/38 (15.7%), 3/7 (42.8%), and 15/51 (29.4%), respectively).

## 4. Discussion

The prenatal and postnatal data presented in this study are among the largest reported series of orofacial clefts diagnosed in a low-risk population.

The overall accuracy of prenatal diagnosis of iCL and/or psCP found in our study was low. The diagnostic accuracy was related to the type of cleft. When iCL was involved, the accuracy of diagnosis was higher and the diagnosis occurred earlier in the pregnancy. These results reflect previous observations that prenatal sonographic detection of iCL is high, whereas the prenatal diagnosis of psCP is much more challenging. A recent systemic review by Divya et al. showed that the accuracy of ultrasound diagnosis of clefts varied widely and has increased over the years with improvements of ultrasound equipment [21]. Furthermore, they stated that a combination of 2D and 3D ultrasound scans have the same accuracy for cleft lip. However, if a cleft lip is suspected, 3D ultrasound should be used for secondary evaluation after the 2D ultrasound and for better evaluation of the fetal palate [21].

The diagnosis of cleft lip and palate in the first trimester has been reported infrequently in the literature. In the current study, most cases involving iCL were diagnosed during the early anomaly scan performed at 14–16 weeks of gestation. This reflects the good visualization of the fetal lips due to advanced technology and a highly experienced technician that enables diagnosis at this stage. However, early prenatal diagnosis of psCP involvement is much more complicated. As described by Burnell et al. [22], when the lips were normal and the cleft involved the palate only, the diagnosis was more difficult. Most cases of psCP alone reported here were diagnosed during the second and third trimesters.

The presence of anomalies in addition to iCL and/or psCP has enormous significance on the outcome and prognosis of these cases. Monlleo et al. reported that associated defects or syndromes were found in 93% and 59.5% of their cohort, respectively. Cleft palate was statistically associated with a greater number of minor defects (*p* < 0.0012) and syndromes [23]. Maarse et al. reported that in 20 prenatal and postnatal studies on associated anomalies and chromosomal defects in clefts, non-isolated iCL/psCP is more frequently associated with chromosomal abnormalities, which are found in approximately 50.7% of cases compared with 0.9% of cases with isolated cleft [24].

The increasing use of prenatal chromosomal microarray analysis (CMA) enables us to detect additional associated genetic abnormalities that would have been missed by using karyotype only. Jin et al. reported that CMA is a valuable tool for identifying submicroscopic chromosomal abnormalities in the prenatal diagnosis of oral clefts [25]. They found that compared with traditional karyotyping, CMA has superior sensitivity (7.7% vs. 3.6%) [25]. Hence, CMA should be part of the routine work-up in cases of prenatal diagnosis of facial cleft.

Most cases of orofacial clefts in our study were isolated, without additional anomalies. In the non-isolated cases, we found significantly higher rates of respiratory, swallowing, and speech complications, as well as a need for psychological therapy. This finding has enormous impact on the prognosis of these patients, as well as on the prenatal counseling given to parents. We assume that this information will give the parents more confidence and reassurance regarding the post-natal treatment, sequelae, and complications of the prenatal diagnosis.

We did not find more cases of associated genetic abnormalities in fetuses with iCL/psCP. Among 22 cases of non-isolated iCL/psCP, only 6 had CMA analysis and among 74 non-isolated CL/CP, 16 had CMA analysis. All tests yielded normal results. In part, this may be due to the low genetic testing rate.

The emotional impact on the parents presents a dilemma regarding possible termination of pregnancy and their child’s prognosis after birth [26,27]. However, the possibility of termination or early preparation is not always possible because the diagnosis may be made late in the pregnancy or postpartum. In our cohort, 59% of the cases of iCL/psCP were diagnosed only after birth. However, postnatal parental counseling is also very important.

Parental counseling of orofacial cleft diagnosis involves discussing the treatment the child is likely to need and encompasses the child’s entire physical and psychological wellbeing. It includes surgical, cosmetic, speech, and psychological treatments since all of these are major components of the long-term sequelae.

Children diagnosed at birth had more swallowing problems (63.2% vs. 43.6%, *p* = 0.04) and fewer surgeries in the first year of life (42.1% vs. 94.9%, *p* < 0.0001), compared with children diagnosed prenatally. They also had more psychological problems (42.1% vs. 17.9%, *p* = 0.01). This can be related to later diagnosis and dealing with early symptoms with a yet unknown cause, or perhaps prenatal diagnosis allows more time for treatment planning and coordination.

Early multidisciplinary evaluation and long-term follow-up of patients with orofacial clefts are essential to achieving optimal clinical outcomes. The growing availability of prenatal diagnosis should also promote multidisciplinary assessment and consultation.

### Strength and Limitation

Because of the nature of this study, we cannot discuss pregnancy outcomes regarding termination and intrauterine fetal demise. Strengths of this study are that it included a large, well-defined cohort. All cases were diagnosed and followed by a pediatric ENT in a university-affiliated tertiary medical center. In addition, this is among the largest reported data series of fetal iCL and/or psCP with postnatal follow-up and prenatal analysis. Data retrieval from patient files was meticulous and we were able to obtain comprehensive information regarding the prenatal work-up and postnatal outcomes of the cases.

Some limitations were related to the lack of uniform imaging throughout the 20-year period of the study. Other limitations were derived from the retrospective nature of the study, including potential selection bias due to patients lost to follow-up. Another issue is recall bias, because the study follow-up was retrospective and based on questionnaire data.

## 5. Conclusions

In conclusion, the accuracy rate of prenatal diagnosis is low. Our results better specify the essential work-up of fetuses with iCL and/or psCP and should improve the parental counseling provided, as well as their understanding and compliance regarding the post-natal therapeutic plan.

## Figures and Tables

**Figure 1 children-09-01880-f001:**
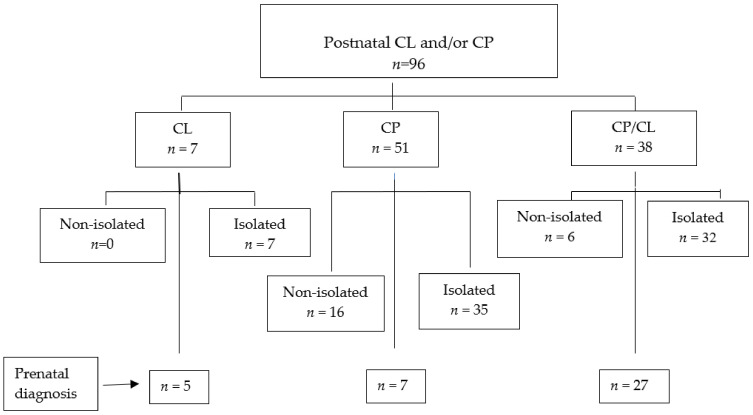
Flow diagram of the study group.

**Table 1 children-09-01880-t001:** Clinical characteristics of the study group (*n* = 96).

Characteristics	*n* (%)
General	
Maternal age at diagnosis, years ± SD	29.9 ± 5.5
Gravity ± SD	3.00 ± 1.9
Parity ± SD	2.59 ± 1.7
Family history	
Parent with iCL/psCP	8 (8.3)
Sibling with iCL/psCP	6 (6.25)
Prenatal diagnosis	39 (40.6)
Prenatal diagnosis—Type of cleft	
iCL	5 (5.2)
psCP	7 (7.3)
iCL ± psCP	27 (28.1)
Male	54 (56.2)
Female	42 (43.8)
Twin pregnancy	7 (7.2)
Chorionic villus sampling or amniocentesis	32 (33.3)
Fetal chromosomal microarray analysis	22 (22.9)
Prenatal otolaryngology consultation	15 (15.6)
Other anomalies at prenatal diagnosis	
Cardiac	2 (2)
CNS	2 (2)
Urinary	1 (1)
Multiple systems	6 (6.25)
Postnatal Diagnosis	
Cardiac	2 (2)
CNS	1 (1)
Urinary	2 (2)
Other facial anomalies	5 (5.2)
Multiple systems	1 (1)
Mode of delivery	
Normal vaginal delivery	71 (73.9)
Operative vaginal delivery	5 (5.2)
Cesarean section	20 (20.8)
Gestational age at delivery, weeks ± SD	38.9 ± 1.8
Postpartum diagnosis	
iCL	7 (7.2)
psCP	51(53)
iCL ± psCP	38 (39.5)
Postpartum diagnosis of additional anomalies	135 (13.5)
Birth weight (g ± SD)	3149 ± 602

iCL, incomplete cleft lip; psCP, cleft of the primary palate and/or the secondary palate.

**Table 2 children-09-01880-t002:** Demographic, obstetric, work-up and outcome of fetuses with postpartum diagnosis of iCL versus psCP and combined iCL-psCP (*n* = 96).

Type of Malformation	iCL *n* (%)	psCP *n* (%)	iCL ± psCP *n* (%)	Adjusted *p*-Value
Postpartum diagnosis	7 (7.2)	51 (53.1)	38 (39.5)	0.003
Prenatal ultrasound diagnosis	5 (71)	7 (13.8)	27 (71)	0.001
Prenatal diagnosis timing (weeks)				
14–20	3 (42.8)	1 (2)	15 (39.5)	0.005
21–24	2 (28.6)	3 (5.9)	8 (21)	
>24	0	3 (5.9)	4 (10.5)	
Male fetus	3 (42.8)	25 (49)	25 (65.7)	0.29
Chorionic villus sampling or amniocentesis	4 (57.1)	14 (27.4)	14 (36.8)	0.25
Fetal chromosomal microarray analysis	3 (42.8)	9 (17.6)	10 (26.3)	0.27
Fetal magnetic resonance imaging	0	4 (7.8)	3 (7.89)	0.24
Prenatal otolaryngology consultation	0	3 (5.9)	12 (31.5)	0.005
Other anomalies at diagnosis *	0	16 (31)	6 (15.7)	0.003
Gestational age at delivery, weeks ± SD	39.43 ± 1.5	38.90 ± 1.0	39.0 3 ± 2.3	0.77
Apgar score ≤ 7	0	3 (5.9)	0	0.93
Postnatal care: respiratory complications requiring intervention	0	10 (19)	5 (13)	0.04
Complications before the first year of life				
Respiratory	0	14 (27)	2 (5)	0.01
Swallowing	1 (14)	35 (68)	17 (44.7)	0.006
Speech	1 (14.2)	19 (37)	5 (13)	0.03
Sequelae				
Respiratory	0	10 (19.6)	1 (12.5)	0.02
Swallowing	1(14.2)	17 (33.3)	2 (5.2)	0.004
Speech	2 (28.5)	41(80)	20 (52.6)	0.001
Surgical interventions before the first year of life	5 (71.4)	20 (39)	36 (94.7)	0.005
Surgical interventions after the first year of life	4 (57.1)	45 (88.2)	23 (60.5)	0.004
Cosmetic interventions	3 (42.8)	2 (3.9)	10 (26.3)	0.002
Speech therapy	3 (42.8)	41 (80.3)	23 (60.5)	0.001
Emotional therapy	3 (42.8)	15 (29.4)	6 (15.7)	0.0002

* Other anomalies at diagnosis included musculoskeletal, cardiac, CNS, urinary system, GI system, facial.

**Table 3 children-09-01880-t003:** Demographic, obstetric, work-up, and outcome of newborns with isolated iCL/psCP versus iCL/psCP associated with additional abnormalities.

Characteristics	Isolated iCL/psCP (*n* = 74)	iCL/psCP with Additional Abnormalities (*n* = 22)	*p*-Value
General			
Maternal age at diagnosis (range)	29.75 ± 5.3	30.02 ± 6.2	0.9
Gravity	2.80 ± 1.8	4.31 ± 2.3	0.21
Parity	2.40 ± 1.6	3.85 ± 2.0	0.2
Family history			
Parent with iCL/psCP	6/74 (%)	2/22 (%)	1
Sibling with iCL/psCP	4/74 (%)	4/22 (%)	0.62
Male fetus	42/74 (56.8%)	12/22 (54.5%)	0.85
Genetic testing			
Fetal karyotype	8/74 (10.8%)	2/22 (9%)	0.73
Fetal chromosomal microarray analysis	16/74 (21.6%)	6/22 (27.2%)	0.58
Fetal magnetic resonance imaging	3/74 (4.1%)	4/22 (18.2%)	0.046
Prenatal otolaryngology consultation	12/74 (16.2%)	3/22 (13.6%)	1
Other anomalies at diagnosis *	0	16 (31)	6 (15.7)
Gestational age at delivery (range)	38.99 ± 1.9	39.00 ± 1.8	0.97
Apgar score ≤ 7	0/74	3/22 (13.63%)	<0.001
Postnatal care			
Medical conditions before the first year of life			
Respiratory complications	6/74 (8.1%)	10/22 (45.5%)	<0.001
Swallowing complications	38/74 (51.4%)	15/22 (68.2%)	0.005
Speech complications	14/74 (18.9%)	11/22 (50%)	0.011
Sequelae			
Respiratory sequelae	4/74 (5.4%)	7/22 (31.8%)	0.003
Swallowing	10/74 (13.5%)	10/22 (45.5%)	0.005
Speech	43/74 (58.1%)	20/22 (90.9%)	0.011
Surgical interventions before the first year of life	49/73 (67.1%)	12/22 (54.5%)	0.281
Surgical interventions after the first year of life	53/74 (71.6%)	19/22 (86.4%)	0.303
Cosmetic	14/74 (18.9%)	1/22 (4.5%)	0.265
Speech therapy	49/74 (66.2%)	1/22 (4.5%)	0.319
Psychological therapy	19/74 (25.7%)	12/22 (54.5%)	0.035

* Other anomalies at diagnosis include musculoskeletal, cardiac, CNS, urinary system, GI system, facial.

## Data Availability

Data are available upon request.

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
