# Peer review of "Cleft Lip and/or Cleft Palate: Prenatal Accuracy, Postnatal Course, and Long-Term Outcomes"

_children, 2022, doi:10.3390/children9121880_

Round 1
Reviewer 1 Report
Introduction. When writing about importance of prenatal diagnosis importance of feeding problems and parent's preparation to solve them should be mentioned (special feeding bottles, specialist consultations, even long term observation at home environment if required), articles could be cited. Breathing and feeding are main issues to solve.
It is not clear what means cosmetic surgery and cosmetic treatment, should be explained. Nowadays It could not be primary cleft lip reconstruction but some scar revision ?
Results.
Table 4. To compare outcomes of CL/CP diagnosed during pregnancy vs. CL/CP diagnosed after pregnancy is not correct. Variables compared are related to types of cleft but not when diagnosis is made (except psychological problems). Respiratory, swallowing and speech problems are characteristic to CP with or without CL (literature and results of this study presented in Table 2). However, as results showed CP was not diagnosed prenataly with the same accuracy as in case of CL with or without CP. Surgical interventions both timing and numbers are related to types of clefts as well, and compare such variables between different types of clefts is not correct (it is comparison of apples and pears). This aspect should be reconsidered in data in table 2 as well. To me to present results in such manner is not correct and guiding to misunderstandings. I firmly recommend to delete Table 4 content except variable - psychological problems, that could be described in text. This part in discussion should be revised too (rows 266-269). To remove from table 2 comparison of surgical interventions and cosmetic interventions between different types of clefts.
Author Response
Reviewer #1: 1. Introduction. When writing about importance of prenatal diagnosis importance of feeding problems and parent's preparation to solve them should be mentioned (special feeding bottles, specialist consultations, even long term observation at home environment if required), articles could be cited. Breathing and feeding are main issues to solve. Author's reply: Thank you for your comment. We added the following sentence at the introduction section. " At the beginning parents might meet with breathing and feeding difficulties which need to be addressed and dealt with (special feeding bottles, specialist consultations for right positioning, alternative feeding methods, etc)". 2. It is not clear what means cosmetic surgery and cosmetic treatment, should be explained. Nowadays It could not be primary cleft lip reconstruction but some scar revision? Author's reply: Thank you for your comment. We added "scar revision" as an explanation for the cosmetic treatments. 3. Results. Table 4. To compare outcomes of CL/CP diagnosed during pregnancy vs. CL/CP diagnosed after pregnancy is not correct. Variables compared are related to types of cleft but not when diagnosis is made (except psychological problems). Respiratory, swallowing and speech problems are characteristic to CP with or without CL (literature and results of this study presented in Table 2). However, as results showed CP was not diagnosed prenataly with the same accuracy as in case of CL with or without CP. Surgical interventions both timing and numbers are related to types of clefts as well, and compare such variables between different types of clefts is not correct (it is comparison of apples and pears). This aspect should be reconsidered in data in table 2 as well. To me to present results in such manner is not correct and guiding to misunderstandings. I firmly recommend to delete Table 4 content except variable - psychological problems, that could be described in text. This part in discussion should be revised too (rows 266-269). To remove from table 2 comparison of surgical interventions and cosmetic interventions between different types of clefts. Author's reply: Thank you for your important comment. The table was deleted.Reviewer 2 Report
The authors present a nice study on the success of ultrasonography diagnosis of orofacial cleft at different stages of fetal development / gestational ages as well as incorporating postnatal follow up, longer term outcomes and data on supplemental genome analysis. Overall this is a useful study that will be a valuable contribution to the literature.
There are however numerous issues that need to be addressed. Many of these are fairly minor and can be fixed with some rewording or additional explanation. There is also one larger concern that really must be corrected and which may affect the entire analyses (although likely no the overall message). So although I feel this is necessary to fix, I do not believe it will impact the importance of the study but may impact some of the numbers that underpin the interpretations. This issue relates to the unusual definition of the abbreviation CP. In this manuscript the authors define CL using Millard's revision of the original Kernahan classification system of orofacial clefts - specifically clefts involving fields 2 and/or 6. This is a simple classification but is satisfactory and reasonable for the purposes of this study. However, the typical and widely used definition of the abbreviation "CP" is for referring to clefts of the secondary palate only. In this paper, the authors use it to mean clefts of any part of the alveolus, primary and secondary palate. As a result, this will create significant confusing and ultimately further chaos within the literature. While they have done the correct thing and defined what they are including under the term "CP", unfortunately many readers will not appreciate the difference. In addition, both biologically and clinically, clefting of the alveolus+primary palate and distinct from clefts of just the secondary palate and are treated very differently in terms of research and clinical management/surgery. Consequently, this will add further confusion. Traditionally, clefts of the lip proper and clefts of the alveolus+primary palate are considered part of the same spectrum (that all have their origins in a defect during the sixth week of development). Secondary palate clefts are common in individuals with lip/alveolus/primary palate clefts and this can have a primary causal mechanism but can also be due to a secondary (indirect) impact. Isolated cleft palate typically refers to clefts of the secondary palate only (and can have involvement of the hard or soft palate components or both) and can arise due to a primary defect in palatal tissue or arise secondary to another disruption in facial development. While the underlying embryology and genetic mechanisms are not directly relevant to the goal of this study, the inclusion of two distinct classes of cleft (and separation of another - or mixing of classes) is indeed problematic. I am sure the authors have this clinical information and so they should group their cleft types appropriately and reassess.
As alluded to earlier, in addition to this major issue described above there are numerous small issues that should be address. These are listed below in the order they appear:
Abstract: The term orofacial cleft includes more than just CL and CP. It also includes rare oblique cleft and other minor associated anomalies. It is suggested that the authors say ‘includes’ not ‘consists of’ as the later suggest that these are the only cleft types under that designation.
Introduction: The authors should be clearer in the use of their cleft terminology. For example, the sentence “A recent study found an increased risk for still broth among fetuses with isolated CL and CP.” Do they mean fetuses with combined CL and CP (ie. CL/P)? Or either isolated CL or isolated CP. It is suggested they introduce the term CL/P as well, specifically when referring to cases with both, to avoid confusion.
The statement “while isolated CL presents mostly as an esthetic issue“ is true ONLY IF CL refers to the lip proper. However, the abbreviation “CL” is typically used to group clefts of the lip proper, alveolus and primary palate – because of the shared developmental origin. When CL includes the alveolus/primary palate then this is not aesthetic. This is why literature consistency is the best option over redefining an abbreviation for one study.
Is there an Ethics protocol number? If so, this should be included for reference.
Any difference in diagnostic rate in those fetuses whose parent or sibs are also affected? i.e. how does having a family member affected influence early diagnosis? Is there more attention paid to checking during the ultrasound because of a family history?
Top of page 5 ; diagnosis was confirmed in 3/7 etc…. when was a confirmation diagnosis being made? At a follow up ultrasound? ENT? Please state. Shouldn’t the confirmed cases be out of the smaller prenatal diagnosis number eg 3/5 (since 5/7 were prenatally diagnosed). You cannot confirm a diagnosis if it was not initially made. I think this is just related to how the authors are using ‘confirmed’. In this case, ‘confirmed’ is being used to infer nothing has changed since the first diagnosis, whereas ‘partial’ diagnosis is referring to a diagnosis that was missed or not made at a later ultrasound. These are not the best words to use. Confirmed or verified may be OK for the first part of this, but perhaps ‘missed’ might be better than ‘partial’.
Did prenatal ultrasound fail to detect the % of cases or was it, more accurately, the ultrasonographer failed to detect these clefts (whether inexperience or lack of resolution of the scanner, or other issues).?
Did those that had a fetal MRI later improve or change diagnosis? What are the authors recommendations around needing or using MRI routinely?
Table 3: superscript “&” and “#” are not explained in the legend.
Table 4 would be more useful if separated out between isolated and non-isolated, pre and postpartum diagnosed. Are the few significant differences just because of an imbalance between isolated and non-isolated in the two groups?
Discussion (line 216) - use either “in” or “of” not both in “with improvements in of ultrasound equipment”
(Line 254) – instead of “We assume this is due to low genetic testing rate.” It might be more accurate or appropriate to say “In part, this may be due to the low genetic testing rate.”
(Line 256) – remove the comma after “parents”
(Line 259) – Change ‘out’ to ‘our’ in the sentence “In out cohort, 59% of the cases of CL/CP were diagnosed only after birth.” Also, it would be appropriate to perhaps speculate on the reason why the diagnosed percentage decreases in the older fetuses when one would expect it might be simpler or more reliable in a larger fetus.
(Line 266-267) – The first sentence is an odd statement. Diagnosis of orofacial clefting is typically made AT BIRTH not sometime during the first year – so some rewording is required. The authors should posit why those diagnosed ‘during the first year of life’ - ie. postnatally – would have fewer surgeries. Is it simply, prenatal diagnosis allows more time for treatment planning and coordination with in a team?
Author Response
Reviewer #2: 1. The comment regarding the classification system of orofacial clefts: Author's reply: We added a paragraph in the methos section: "The diagnosis was based on physical examination and imaging, and was classified using Millard classification: class 2 and 6, which involve only the soft tissue, were considered CL (known also as "incomplete primary cleft palate/premaxilla") , while classifications 3, 4 and 7-11, which involve the hard primary palate and the secondary palate, were considered CP [3]. We preferred in this paper this classification over the embryonal (primary vs. secondary) because it separates the clinical severity to mainly aesthetic with CL as opposed to mainly functional with CP". 2. Abstract: The term orofacial cleft includes more than just CL and CP. It also includes rare oblique cleft and other minor associated anomalies. It is suggested that the authors say ‘includes’ not ‘consists of’ as the later suggest that these are the only cleft types under that designation. Author's reply: The word "consist" was changed to "include". 3. Introduction: The authors should be clearer in the use of their cleft terminology. For example, the sentence “A recent study found an increased risk for still broth among fetuses with isolated CL and CP.” Do they mean fetuses with combined CL and CP (ie. CL/P)? Or either isolated CL or isolated CP. It is suggested they introduce the term CL/P as well, specifically when referring to cases with both, to avoid confusion. Author's reply: Thank you for the important comment. The change was made. 4. The statement “while isolated CL presents mostly as an esthetic issue“ is true ONLY IF CL refers to the lip proper. However, the abbreviation “CL” is typically used to group clefts of the lip proper, alveolus and primary palate – because of the shared developmental origin. When CL includes the alveolus/primary palate then this is not aesthetic. This is why literature consistency is the best option over redefining an abbreviation for one study. Author's reply: The following change was made: Primary CP may involve only the lip (CL), which presents mostly as an esthetic issue, or may involve the alveolar ridge and the palate anterior to the incisive foramen which is then associated with dental eruption problems and orthodontics [19]. 5. Is there an Ethics protocol number? If so, this should be included for reference. Author's reply: Yes. The protocol number was added. 6. Any difference in diagnostic rate in those fetuses whose parent or sibs are also affected? i.e. how does having a family member affected influence early diagnosis? Is there more attention paid to checking during the ultrasound because of a family history? 7. Author's reply: Thank you for the important question. We asked for family history when data was collected (Table 1). However, unfortunately, we don't have the data regarding the diagnostic rate in this cases compared to cases with no family history. 8. Top of page 5 ; diagnosis was confirmed in 3/7 etc…. when was a confirmation diagnosis being made? At a follow up ultrasound? ENT? Please state. Shouldn’t the confirmed cases be out of the smaller prenatal diagnosis number eg 3/5 (since 5/7 were prenatally diagnosed). You cannot confirm a diagnosis if it was not initially made. I think this is just related to how the authors are using ‘confirmed’. In this case, ‘confirmed’ is being used to infer nothing has changed since the first diagnosis, whereas ‘partial’ diagnosis is referring to a diagnosis that was missed or not made at a later ultrasound. These are not the best words to use. Confirmed or verified may be OK for the first part of this, but perhaps ‘missed’ might be better than ‘partial’. Author's reply: The word "confirmed" was changed to "verified". The word "partial" was changed to "missed". 9. Did prenatal ultrasound fail to detect the % of cases or was it, more accurately, the ultrasonographer failed to detect these clefts (whether inexperience or lack of resolution of the scanner, or other issues).? Author's reply: This is an important question. There is always the possibility that the diagnosis was missed by the sonographer. However, all anatomic scans were performed by experienced experts in obstetric ultrasound. 10. Did those that had a fetal MRI later improve or change diagnosis? What are the authors recommendations around needing or using MRI routinely? Author's reply: Thank for the interesting question. Prenatal fetal magnetic resonance imaging was performed only in 7 (17.9%) cases where the diagnosis of CP was uncertain. It is a small number and conclusions cannot have been made. 11. Table 3: superscript “&” and “#” are not explained in the legend. Authors's reply: Corrected. 12. Table 4 would be more useful if separated out between isolated and non-isolated, pre and postpartum diagnosed. Are the few significant differences just because of an imbalance between isolated and non-isolated in the two groups? Author's reply: Table 4 was deleted. 13. Discussion (line 216) - use either “in” or “of” not both in “with improvements in of ultrasound equipment” Author's reply: Corrected.Round 2
Reviewer 2 Report
I thank the authors for their responses. The manuscript reads more smoothly as a result and helps clarify the various minor issues that were raised. That said, the one main issues still persists even though I can appreciate the argument being made by the authors and why this grouping is appropriate for their study. However, it does not mitigate the confusion that will inevitably arise if not changed. Consequently, I have proposed a simple solution that I hope the authors will consider. In addition, I have picked up one or two additional minor errors that have crept in as a result of revising the manuscript.
New minor issue:
The new sentence in the Introduction beginning “At the beginning parents might meet with breathing and feeding difficulties….” Does not read well as it makes these issues sound like they are parental issues. But these are patient issues that the parents need to address. I would suggest rewording the start of this sentence to: “From birth, the cleft patient may face breathing and feeding difficulties which need…..”
New minor issue:
The new sentence “Primary CP may involve only the lip (CL), which presents mostly as an esthetic issue, or may involve the alveolar ridge and the palate anterior to the incisive foramen which is then associated with dental eruption problems and orthodontics [19].” is mostly helpful. However, the use now of ‘Primary CP may involve only the lip (CL)” has not helped with the confusion. The authors argue that CL is used in this study to include lip and alveolar clefts but not involving the primary palate. So, their statement above is incomplete and again misleading.
Instead of saying “Primary CP”, I would suggest tying this instead to the embryological defect - for example:
“Disruption of the fusion of the embryonic facial tissues that form the lip, alveolus and primary palate may present as a cleft of the lip only, the alveolus only or more commonly both together. The alveolar cleft often will continue either side of the primary palate to the incisive foramen. These presentations are collectively abbreviated as CL. Such presentations are typically associated with dental eruption problems necessitating orthodontic treatment [19].”
Primary issue to be addressed:
In the Methods, the authors have added some new sentences to help with clarifying for the reader how they have grouped cases. “The diagnosis was based on physical examination and imaging, and was classified using Millard classification: class 2 and 6, which involve only the soft tissue, were considered CL (known also as "incomplete primary cleft palate/premaxilla") , while classifications 3, 4 and 7-11, which involve the hard primary palate and the secondary palate, were considered CP [3]. We preferred in this paper this classification over the embryonal (primary vs. secondary) because it separates the clinical severity to mainly aesthetic with CL as opposed to mainly functional with CP.” This is appreciated and from the point of view of the ultrasonographer (ie. this study), there is a reasonable argument for separating clefting differently to the embryological separations. And I do not have an issue with doing this separation for this study – and the authors justify this well. However, the authors MUST AVOID using the abbreviations CL and CP, because they are not the same CL and CP that they talk about in the Introduction. As a possible solution that I would find satisfactory/acceptable, the authors could simply use unique/new abbreviations. eg. iCL (incomplete CL) and psCP (primary+secondary cleft palate).
If the above solution was adopted, then the newly added section might read something like: “The diagnosis was based on physical examination and imaging, and was classified using the Millard classification system into two distinct groups. The first group included class 2 and class 6 clefts, which involve only the soft tissue. These were considered incomplete clefts (abbreviation hereafter as iCL). The second group included classes 3, 4 and 7-11, which all involve the hard primary palate and/or the secondary palate (abbreviated hereafter as psCP) [3]. These groupings were made in preference to the embryonal (primary vs. secondary) classification because it separates the clinical severity to mainly aesthetic with iCL as opposed to mainly functional with psCP, and better relates to the current diagnostic limitations of ultrasonography.”
It would be from this point forward in the manuscript (including the Tables and Figures), that the authors would then need to change from the use of CL and CP to the new abbreviations. The Introduction would remain unchanged but the Abstract would need fixing as it will contain a mix of the traditional use of the abbreviations and the new abbreviations.
New minor issue:
In the Discussion, the authors need to correct the following sentence from “Children diagnosed at birth of life had more swallowing problems…” to “Children diagnosed at birth had more swallowing problems…”
Author Response
Dear Reviewer, Thank you so much for your important and detailed comments. 1. The new sentence in the Introduction beginning “At the beginning parents might meet with breathing and feeding difficulties….” Does not read well as it makes these issues sound like they are parental issues. But these are patient issues that the parents need to address. I would suggest rewording the start of this sentence to: “From birth, the cleft patient may face breathing and feeding difficulties which need…..” Authors reply: Thank you for you comment. The sentence was changed. 2. The new sentence “Primary CP may involve only the lip (CL), which presents mostly as an esthetic issue, or may involve the alveolar ridge and the palate anterior to the incisive foramen which is then associated with dental eruption problems and orthodontics [19].” is mostly helpful. However, the use now of ‘Primary CP may involve only the lip (CL)” has not helped with the confusion. The authors argue that CL is used in this study to include lip and alveolar clefts but not involving the primary palate. So, their statement above is incomplete and again misleading. Instead of saying “Primary CP”, I would suggest tying this instead to the embryological defect - for example: “Disruption of the fusion of the embryonic facial tissues that form the lip, alveolus and primary palate may present as a cleft of the lip only, the alveolus only or more commonly both together. The alveolar cleft often will continue either side of the primary palate to the incisive foramen. These presentations are collectively abbreviated as CL. Such presentations are typically associated with dental eruption problems necessitating orthodontic treatment [19].” Authors reply: Thank you for you comment. The sentence was changed. 3. In the Methods, the authors have added some new sentences to help with clarifying for the reader how they have grouped cases. “The diagnosis was based on physical examination and imaging, and was classified using Millard classification: class 2 and 6, which involve only the soft tissue, were considered CL (known also as "incomplete primary cleft palate/premaxilla") , while classifications 3, 4 and 7-11, which involve the hard primary palate and the secondary palate, were considered CP [3]. We preferred in this paper this classification over the embryonal (primary vs. secondary) because it separates the clinical severity to mainly aesthetic with CL as opposed to mainly functional with CP.” This is appreciated and from the point of view of the ultrasonographer (ie. this study), there is a reasonable argument for separating clefting differently to the embryological separations. And I do not have an issue with doing this separation for this study – and the authors justify this well. However, the authors MUST AVOID using the abbreviations CL and CP, because they are not the same CL and CP that they talk about in the Introduction. As a possible solution that I would find satisfactory/acceptable, the authors could simply use unique/new abbreviations. eg. iCL (incomplete CL) and psCP (primary+secondary cleft palate). Authors reply: Thank you so much for the important an helpful comment. The proposed solution was adopted, and everything was changed accordingly. 4. In the Discussion, the authors need to correct the following sentence from “Children diagnosed at birth of life had more swallowing problems…” to “Children diagnosed at birth had more swallowing problems…” . Author's reply: The sentence was corrected.